# Towards the Validation of Executive Functioning Assessments: A Clinical Study

**DOI:** 10.3390/jcm11237138

**Published:** 2022-11-30

**Authors:** Daniel Faber, Gerrit M. Grosse, Martin Klietz, Susanne Petri, Philipp Schwenkenbecher, Kurt-Wolfram Sühs, Bruno Kopp

**Affiliations:** Department of Neurology, Hannover Medical School, Carl-Neuberg-Straße 1, 30625 Hannover, Germany

**Keywords:** neuropsychological assessment, psychometric theory, validity, intelligence, executive function, Raven’s matrices, vocabulary test, Wisconsin card sorting, verbal fluency, figural fluency

## Abstract

Neuropsychological assessment needs a more profound grounding in psychometric theory. Specifically, psychometrically reliable and valid tools are required, both in patient care and in scientific research. The present study examined convergent and discriminant validity of some of the most popular indicators of executive functioning (EF). A sample of 96 neurological inpatients (aged 18–68 years) completed a battery of standardized cognitive tests (Raven’s matrices, vocabulary test, Wisconsin Card Sorting Test, verbal fluency test, figural fluency test). Convergent validity of indicators of intelligence (Raven’s matrices, vocabulary test) and of indicators of EF (Wisconsin Card Sorting Test, verbal fluency test, figural fluency) were calculated. Discriminant validity of indicators of EF against indicators of intelligence was also calculated. Convergent validity of indicators of intelligence (Raven’s matrices, vocabulary test) was good (*r_xtyt_* = 0.727; *R*^2^ = 0.53). Convergent validity of fluency indicators of EF against executive cognition as indicated by performance on the Wisconsin Card Sorting Test was poor (0.087 ≤ *r_xtyt_* ≤ 0.304; 0.008 ≤ *R*^2^ ≤ 0.092). Discriminant validity of indicators of EF against indicators of intelligence was good (0.106 ≤ *r_xtyt_* ≤ 0.548; 0.011 ≤ *R*^2^ ≤ 0.300). Our conclusions from these data are clear-cut: apparently dissimilar indicators of intelligence converge on general intellectual ability. Apparently dissimilar indicators of EF (mental fluency, executive cognition) do not converge on general executive ability. Executive abilities, although non-unitary, can be reasonably well distinguished from intellectual ability. The present data contribute to the hitherto meager evidence base regarding the validity of popular indicators of EF.

## 1. Introduction

Executive functioning (EF) is a construct of fundamental importance for cognitive neuropsychology, although a definition of which cognitive abilities are denoted by the term EF is not yet available. There seems to be a consensus that EF encompasses ‘higher’ cognitive functions, usually defined as a set of domain-general cognitive control mechanisms supporting goal-directed behavior (e.g., [1]), but their exact nature remains a matter of debate [2,3,4,5].

Many cognitive neuropsychologists share the widely held—yet barely evidence-based—belief that EF represents a cognitive construct that is separable from general intellectual abilities, in particular from intelligence (e.g., [1]). David Wechsler once defined intelligence as the “the global capacity of the individual to act purposefully, to think rationally and to deal effectively with his environment” [6] (p. 3). From Wechsler’s definition, it becomes evident that intelligence and EF may share substantial conceptual overlap.

Psychological science in the 20th century has evidenced a controversy about the most reasonable theoretical model of intelligence [7]. Spearman initially identified a single general intellectual ability, for which he coined the term *g* (for “general factor” [8,9], but see [10]). Meanwhile, a consensus regarding the dimensionality of intelligence has only been achieved insofar as most researchers agree with the assumption that cognitive abilities underlying intelligence are organized in a hierarchical structure, with *g* at its highest level. Cattell [11] distinguished two types of cognitive abilities that are relevant for general intelligence in a revision of Spearman’s concept of *g*. Cattell hypothesized fluid intelligence (*g_f_*) as the ability to solve novel problems by using reasoning, while he hypothesized crystallized intelligence (*g_c_*) as a knowledge-based ability, which is heavily dependent on education. Horn [12] identified a number of additional broad cognitive abilities in a revision of the *g_f_*-*g_c_* theory, and Carroll [13] proposed a hierarchical model of intelligence with three levels, which is now known as the CHC (Cattell–Horn–Carroll) model [14]. The bottom level of the CHC model consists of highly specialized, task-specific cognitive abilities. The middle level of the CHC model consists of Horn’s broad cognitive abilities, including—but not limited to—*g_f_* and *g_c_*. Carroll accepted Spearman’s concept of *g* as representing the highest level of intellectual abilities, but affecting performance on any particular test solely via its influence on the identified broad cognitive abilities such as *g_f_* and *g_c_* [14].

The present study focusses on validating putative indicators of EF. Cronbach once characterized the problem of validity in the following words: “To defend the proposition that a test measures a certain variable defined by a theory, one looks basically for two things. The first is *convergence* of indicators. There need to be two or more different kinds of data that are regarded as suitable evidence that a person is high or low on the variable. If these indicators agree, despite their surface dissimilarity, we place greater faith in the proposed theoretical interpretation. […] The second kind of evidence is *divergence* of indicators that are supposed to represent different constructs. If a test is said to measure “ability to reason with numbers,” it should not rank pupils in the order a test of sheer computation gives, because the computation test cannot reasonably be interpreted as a reasoning test. The test interpretation should also be challenged if the correlation with a test of verbal reasoning is very high, because this would suggest that general reasoning ability accounts for the ranking, so that specialized ability to reason with numbers is an unnecessary concept.” ([15], p. 144; italics in the original text).

Cronbach’s approach to validity was based on two or more different theoretical constructs (which are needed for discriminant validation) with two or more different kinds of data (indicators) per construct (which are needed for convergent validation). The design of the present study, therefore, included two constructs (i.e., intelligence and EF), each of which was represented by two or more indicators. Measures of *g_f_* and *g_c_* were utilized as indicators of intelligence, and measures of executive cognition, also known as cognitive flexibility, verbal and figural fluency, provided indicators of EF. The study aimed at evaluating convergent validity of the named indicators of EF, and it also aimed at evaluating discriminant validity of indicators of EF against indicators of *g_f_* and *g_c_*.

Some intelligence tests target rather directly the assessment of *g_f_* and *g_c_*. The National Adult Reading Test [16] (NART) is often used to assess *g_c_* in clinical neuropsychology, under the assumption that this education-dependent facet of intelligence is relatively insensitive to brain disease and can thus serve as a reasonable indicator of premorbid crystallized intelligence [1]. Raven’s Progressive Matrices [17] (RPM) is often considered as a quintessential indicator of *g_f_* (e.g., [9]). We considered an analogue of the NART (which would not be suitable for German speaking patients) as an indicator of *g_c_*, and a recently standardized variant of the RPM as an indicator of *g_f_*.

The Wisconsin card sorting task [18,19] currently provides one of the most popular assessment techniques for EF [20]. The purpose of the Wisconsin card sorting task is to evaluate the ability to form abstract concepts, to maintain and to shift the mental set in response to verifying or falsifying feedback, respectively. Multiple standardized variants of the Wisconsin card sorting task are now in use in clinical neuropsychology; we prefer the Modified Wisconsin Card Sorting Test (M-WCST; [21]) for reasons that have been outlined elsewhere [22]. M-WCST scores provide three standardized scores, i.e., ‘number of categories correct’, ‘number of perseveration errors’, and their linear combination, which is referred to as ‘executive functioning composite’. These M-WCST scores are thought to provide indicators of essential aspects of executive cognition/cognitive flexibility, namely the ability to abstract (categories) and to remain flexible in response to falsifying feedback (perseveration errors; [20,23]).

Verbal fluency tasks evaluate the spontaneous oral production of words; they have a long history of use in psychology, dating from the work of Thurstone [24]. The most common version of verbal fluency tasks are lexical (a.k.a. letter or phonemic) fluency (here, the task is producing as many words as possible with a specified initial letter) and semantic (a.k.a. category) fluency (here, the task is producing as many words as possible from a specified semantic category). Moderately high correlations between intellectual abilities and verbal fluency have been reported in the literature (for review see [20]).

Design (a.k.a. figural) fluency tasks measure the spontaneous graphical production of novel designs. Design fluency tasks [25] were developed as non-verbal analogs to verbal fluency tasks. Five-point tasks [26] arrange five dots, as on a die, and they request the production of as many unique figures as possible by connecting neighboring dots. Ruff (1987) developed a standardized variant of the five-point task, the Ruff Figural Fluency Test (RFFT; [27,28]). As is the case with verbal fluency, higher intelligence is known to be associated, to some degree, with better figural productivity on the RFFT (for review see [20]).

The relationships between intelligence and EF remain under debate in the neuropsychological literature. Some colleagues have emphasized discriminability of intelligence and EF (e.g., [1,29]), while other authors have claimed that the available indicators of intelligence and of EF merely provide convergent measures of *g* (e.g., [9,30]). More detailed discussions about putative relationships between intelligence and EF can be found in [9,31,32,33].

Attempts towards evidence-based validation are of crucial importance for the further advancement of cognitive neuropsychology [34]. The design of the present study allowed us to to examine multiple validity-related research questions that are relevant for the neuropsychological EF construct. Convergent validity could be examined because each of the two relevant constructs (intelligence, EF) was assessed by multiple indicators (indicators of intelligence included proxies for *g_c_* and *g_f_*; indicators of EF included executive cognition/cognitive flexibility, verbal and figural fluency). Discriminant validity could, likewise, be examined. Discriminant validation of EF against intelligence would be essential, since the EF construct no longer possesses neuroanatomical claims, such as that it represents functions of the frontal lobes. Note that the formerly popular construct of ‘frontal lobe functions’ is no longer conventional in clinical neuropsychology. EF, however, is a purely cognitive construct, without any reference to its potential neuroanatomical substrates. Evaluating the discriminant validity of EF against intelligence is crucial for validating the EF construct. Failures of discriminant validation of EF against intelligence would suggest that EF might be an untenable neuropsychological construct, and that *g* might actually account for inter-individual differences in indicators of EF. Despite the far-reaching implications that validity studies might lead to, discriminant validation of EF against intelligence has been a relatively neglected topic in the literature [35]. Most of the previous validation studies had their methodological grounding in factor-analytic methods e.g., [31,36,37,38,39] and regression-based methods e.g., [40]. Here we deliberately chose an easily applicable, correlative methodology, in order to encourage clinical neuropsychologists to contribute to the evidence-based validation of the EF construct through the proliferation of future studies.

The present study serves to add to the hitherto rather meager evidence base regarding the validation of EF. It is based on analyses of correlative data from a clinical sample of neurological inpatients (see also [29,30]). The patient sample mainly consisted of patients who were referred to our university-based neurological department for diagnosis. Such clinical samples offer the advantage that the full spectrum of cognitive abilities comes under scrutiny, especially when non-selected consecutive samples of patients are studied. Such samples display a huge heterogeneity of individual cognitive abilities across the full ability spectrum, through inclusion of patients with severe and with less severe diseases of the central nervous system, and inclusion of patients who suffer from a peripheral nervous system disease only, along with the inclusion of patients who suffer from a non-neurological disease or no disease.

## 2. Materials and Methods

### 2.1. Participants

We analyzed data that were obtained from 96 consecutively admitted neurological inpatients. A sample size of *n* = 96 is sufficient to determine whether a correlation coefficient *r* = 0.2825 differs from zero (*α* = 0.05 (two-sided); *β* = 0.20; see https://sample-size.net/correlation-sample-size/ accessed on 25 July 2022). The patient sample mainly consisted of patients who were referred to our university-based neurological department for potential neurological diagnosis during the period January to July 2022. They were referred for neuropsychological assessment by the collaborating neurologists (GMG, MK, SP, PS, KWS). Participants had to be between 18 and 69 years old. The choice of this age range allowed the transfer of raw scores to standard scores on all cognitive tests that were conducted. Only patients with German as their native language were included as participants. Exclusion criteria were severe visual or motor dysfunction, dementia, and inadequate vigilance because these symptoms precluded in-depth cognitive testing.

Our sample consisted of 35 male and 60 female patients (plus one patient who preferred not to say). Table 1 summarizes the sociodemographic characteristics of the sample, divided into subsamples of 57 patients who were diagnosed with various brain diseases, and 39 patients without brain diseases. Brain diseases included vascular diseases, autoimmune/inflammatory diseases, and neurodegenerative diseases. Peripheral nervous system diseases (e.g., polyneuropathy, myopathy) and non-neurological (e.g., functional) diseases were subsumed under the second subsample.

### 2.2. Materials and Design

Cognitive testing lasted about 1.5 h per participant. The study design consisted of two intelligence tests (a proxy for fluid intelligence, *g_f_*, and a proxy for crystallized intelligence, *g_c_*), and three tests of EF (executive cognition/cognitive flexibility, verbal fluency, figural fluency), as well as self-report questionnaire regarding non-somatic depressive symptoms. Patients did not receive any additional reward.

#### 2.2.1. Intelligence

##### Fluid Intelligence (*g_f_*): Raven’s Matrices

Raven’s matrices are commonly regarded as a suitable measure of fluid intelligence e.g., [9]. We utilized the recently published German version of Raven’s Progressive Matrices 2 Clinical Edition [41] in the paper-and-pencil format; we refer to this test simply as Raven 2 throughout the article. The Raven 2 consists of five sets of items (Set A–E containing twelve items each, ordered by complexity). Sets A–C are typically conducted with children (aged 4–8 years). However, Sets A–C can also be administered to patients exhibiting intellectual/mental disabilities. We decided to assess Sets A–C with all participants, because our sample comprised individuals with neurological diagnoses. Sets B–E are typically used in the age range 9–69 years. Following successful completion of Sets A–C, we decided whether or not a given patient would continue to complete the remaining Sets D–E. This decision was based upon the patient’s A–C score. Specifically, patients completed Sets D–E only if they achieved at least 18/36 correct items on Sets A–C. Similar procedures were applied in older versions of the Raven’s matrices, namely in the Standard Progressive Matrices [42] and the Colored Progressive Matrices [43]. Time limits were 30 min for Sets A–C (first timer) and 45 min for Sets B–E, respectively. Thus, there were two Raven 2 scores, i.e., number of correct items on Sets A–C (obtained from all participants), and number of correct items on Sets B–E (obtained from only those participants who had at least 18 correct items on Sets A–C). In order to check the time limit for Set B–E (45 min), a second timer always started at the beginning of Set B. The examiner, rather than the examinee, marked the answers on the answer sheet in an attempt to exclude any influence of potential psychomotor slowing/visual–constructive disabilities.

The Raven 2 [41] standardization sample comprised *n* = 1200 healthy individuals across six European countries (*n* = 200 per country). The age range was from 4 to 69 years.

##### Crystallized Intelligence (*g_c_*): Vocabulary

Verbal knowledge is commonly regarded as a suitable measure of crystallized intelligence e.g., [44]. We utilized the Wortschatztest (acronym: WST; literal translation: ‘vocabulary test’, [45]). The WST is a German language vocabulary test and it measures word recognition on 42 rows, each of which is composed of one valid German word embedded in five non-words. The rows are ordered according to difficulty (i.e., higher row numbers contain less frequently utilized words), and subjects have to mark the recognized word. Guessing was discouraged. There was no time limit for task completion. The WST score simply reflects the number of correctly identified words.

The WST standardization sample comprised *n* = 573 healthy participants (*M* = 40 years of age). The age range was from 16 to 90 years.

#### 2.2.2. Executive Functioning

##### Executive Cognition/Cognitive Flexibility: Wisconsin Card Sorting Task

The Modified Wisconsin Card Sorting Test (acronym: M-WCST; [21]) is a commercially available version of the Wisconsin card sorting task. The M-WCST consists of four stimulus cards, which are placed in front of the subject. They depict a red triangle, two green stars, three yellow crosses, and four blue circles, respectively. The subject receives 48 response cards (=48 trials), which can be categorized according to their color, shape, or number. The subject is asked to match each of the 48 response cards to one of the four stimulus cards. After each trial, verbal feedback is given by the examiner (‘correct’ or ‘incorrect’). After six consecutive correct card sorts, the task rules change. The exact test administration followed the arrangements in [22]. To analyze M-WCST performance, we utilized three scores, i.e., number of correct categories (i.e., six consecutive correct rule matches), number of perseveration errors (i.e., rule repetitions following negative feedback), and their linear combination, referred to as ‘executive function composite’ in the manual.

The M-WCST [21] standardization sample comprised *n* = 323 healthy participants (*M* = 54.69 years of age) recruited through random sampling. The age range was from 18 to >85 years.

##### Verbal Fluency

Verbal fluency was assessed with the Regensburger Wortflüssigkeits-Test (acronym: RWT; literal translation: ‘word fluency test’; [46]). Four two minute sub-tests of the RTW were selected, namely lexical fluency (producing words with initial letter S), lexical switching (producing words with initial letters G and R in alternating order), semantic fluency (producing words from the semantic category animals), and semantic switching (producing words from the semantic categories sports and fruits in alternating order). RWT scores simply reflect the number of valid words produced on the sub-tests lexical fluency, lexical switching, semantic fluency, and semantic switching.

The RWT standardization sample comprised *n* = 884 participants, classified into healthy adults (*n* = 634, age range 18 to 65 years), children (*n* = 184, age range 8 to 15 years) and patients (neurological, psychiatric; *n* = 66).

##### Figural Fluency

Figural fluency was assessed with the German version of the Ruff Figural Fluency Test (acronym: RFFT; [47]). The RFFT consists of five parts, each consisting of a page with 35 five-dot patterns. Subjects have to produce as many unique designs as possible by connecting two or more of the five dots in unique ways. The time limit for each of the five parts was one minute and subjects were instructed to avoid design repetitions. If a subject repeated a design on one of the pages, this was considered as a perseverative error. The scores were the sum of unique designs produced across all five parts. An error ratio was also calculated by dividing the number of perseverative errors by the number of unique designs.

The RFFT standardization sample comprised *n* = 358 healthy participants. The age range was from 16 to 70 years.

#### 2.2.3. Depressive Mood

The German version of the Beck Depression Inventory–Fast Screen was used (acronym: BDI-FS, [48]). The BDI–FS is a self-report questionnaire that consists of seven items, which stem from the full 21 item Beck Depression Inventory [49]. The BDI–FS items are intended to measure non-somatic depressive symptoms such as feelings of sadness, pessimism, failure in the past, loss of pleasure, self-dislike, self-criticism, and suicidal thoughts. The score was simply the sum of the single items scores (ranging from 0 to 21).

The BDI–FS [48] provides no standardization sample, but rather displays multiple studies conducted to estimate reliability and validity for this self-report instrument.

### 2.3. Reliability Estimates

The available reliability estimates of all test scores that were considered in the present study are presented in Table 2. Consistency reliability estimates (split-half reliability (*r*_SB_) or Cronbach’s α were preferred when available, and test–retest reliability estimates (*r*_tt_) were utilized otherwise. As a note of caution, the reader is reminded that reliability generalization should not be taken for granted, especially when reliability estimates are transferred from studies of healthy participants to clinical samples [22]. Therefore, the available reliability estimates can only be considered as approximations, and uncertainty regarding the actual reliability of the considered scores must be acknowledged.

#### 2.3.1. Intelligence Reliability

##### Raven 2 Reliability

The Raven 2 manual [41] provides estimates of *r*_SB_ for both Sets, i.e., A–C and B–E, with 0.80 < *r*_SB_ < 0.90, which would typically be considered as reasonably good internal consistencies. However, it should be kept in mind that an influential psychometric textbook recommended that “a reliability of 0.90 is the bare minimum, and a reliability of 0.95 should be considered the desirable standard” [50]. Viewed from the psychometric perspective, Set A–C should be preferred over Set B–E.

##### WST Reliability

The WST manual [45] reports an estimate of split-half reliability that exactly matches the ‘desirable standard’ (i.e., *r*_SB_ = 0.95; [50]).

#### 2.3.2. Executive Functioning Reliability

##### M-WCST Reliability

Kopp et al. [22] reported split-half reliability estimates (*r*_SB_) based on a clinical sample of neurological inpatients (*n* = 146). The M-WCST EFc achieved an estimate of split-half reliability that exactly matches the ‘desirable standard’ [50], while M-WCST categories and M-WCST’ perseverative errors still surpassed the ‘bare minimum’ criterion (i.e., *r*_SB_ > 0.90, [50]).

##### RWT Reliability

We identified one study [51] that reported consistency reliability (Cronbach’s α = 0.90) for RWT lexical fluency, but not for the remaining RWT fluency scores. Regarding the remaining RWT fluency scores, we had to rely on the RWT manual [46] that provides test–retest reliability estimates (*r*_tt_) from a sample of *n* = 90 university students (*M* = 22 years of age) across a relatively short test–retest interval (three weeks). Test–retest reliability estimates ranged between 0.72 ≤ *r*_tt_ ≤ 0.85, akin to results obtained from related reliability studies in English language speaking countries [52,53]. Taken together, the two RWT fluency (lexical, semantic) scores achieved somewhat higher reliability estimates, 0.85 ≤ *rel* ≤ 0.90, compared to the two RWT switching (lexical, semantic) scores, 0.72 ≤ *r*_tt_ ≤ 0.77.

##### RFFT Reliability

The RFFT manual [47] does not present estimates of consistency reliability. Ruff et al. [28] utilized a relatively small sub-sample (*n* = 95) of his original standardization sample, and he then provided test–retest reliability estimates (*r*_tt_) across a relatively long test–retest interval (six months). Fernandez et al. [54] provided an estimate of consistency reliability for the Unique Designs score from the original Five-Point Test [26] based on a non-clinical sample of healthy participants (*n* = 209), with *r*_SB_ = 0.80. Here, we assume that the consistency reliability reported for Five-Point Test UD scores may be generalizable to RFFT UD scores. The consistency reliability of the RFFT ER score remains obscure. One study [55] examined its test–retest reliability based on a sample of *n* = 90 college undergraduates with a somewhat disappointing result, i.e., *r*_tt_ = 0.64.

#### 2.3.3. BDI-FS Reliability

Poole et al. [56] examined the consistency reliability (Cronbach’s α) of the BDI–FS in a sample of *n* = 1.227 chronic pain patients.

### 2.4. The Importance of Reliability

At this point, the distinction between measured scores (observed scores) and their respective true scores in term of classical psychometric test theory [50] comes into the play. Basically, the almost exclusively less than perfect measurement reliability of psychological measures attenuates correlations between observed scores compared to correlations between true scores [57,58]. As an example, consider a correlation between two observed measures (e.g., *r_xy_* = 0.50), and (consistency) reliabilities of *r_xx_* = 0.80 and *r_yy_* = 0.70. Through application of Spearman’s disattenuation equation (Equation (1)), we obtain *r_xtyt_* = 0.67, i.e., a considerably higher correlation between the true score variables *x_t_* and *y_t_* compared to the correlation between the observed score variables *x* and *y*, simply through consideration of their less than perfect measurement reliability.
(1)$$r_{xtyt}=\frac{r_{xy}}{\sqrt{r_{xx}\times r_{yy}}}$$

### 2.5. Two Short Notes on Standard Scores

#### 2.5.1. Rationale for Preferring Standard Scores over Raw Scores

Throughout this article, we considered standard scores rather than raw scores. Consideration of standard scores (such as *z*-scores; Gauss distribution with central tendency *M* = 0 and variability *SD* = 1) have some virtues compared to raw scores. Most importantly, a standard score reveals individual abilities *after* accounting for sociodemographic variables such as age, education, and sex, which are known to exert strong effects on cognitive abilities. As an example, imagine a raw score of 18 out of 36 on Raven 2 A–C. This raw score would probably indicate low individual abilities if obtained from a young, well-educated female, whereas the same raw score would probably indicate higher individual abilities if obtained from an old, poorly-educated male. Thus, standard scores offer potentially age-, education- and sex-adjusted estimates of individual abilities, through adequate socio-economic stratification of sufficiently large standardization samples. This advantage of standard scores, however, does not come without potential caveats, especially when the standardization of the considered assessment instruments follows very heterogeneous procedures (in terms of sample size, socio-economic stratification, etc.).

#### 2.5.2. Resolution Limitations of Standard Scores

Another typical problem of standard scores is their limited resolution. As an example, a test manual may reveal that a particular raw score falls below the one percent percentile, without exact resolution (i.e., the observed standard score equals ‘less than’ one percentile). We applied the following interpolation equation (Equation (2)) in such cases:

‘Less than’ interpolation rule (<percentile):(2)$$\mathrm{interpolated}\;\mathrm{percentile}=\frac{0+‘\mathrm{less}\;\mathrm{than}’\;\mathrm{percentile}}{2}\;$$

‘More than’ interpolation rule (>percentile):$$\mathrm{interpolated}\;\mathrm{percentile}=\frac{100+‘\mathrm{more}\;\mathrm{than}’\;\mathrm{percentile}}{2}$$

The ‘less than’ interpolation described by Equation (2) had to be applied 51 times, as follows: 47 times for ‘percentile < 1’ (interpolated percentile = 0.5), two times for ‘percentile < 1.1’ (interpolated percentile = 0.55), and two times for ‘percentile < 2’ (interpolated percentile = 1). The ‘more than’ interpolation described by Equation (2) had to be applied 15 times, as follows: Once for ‘percentile > 73’ (interpolated percentile = 86.5), once for ‘percentile > 81’ (interpolated percentile = 90.5), seven times for ‘percentile > 88.8’ (interpolated percentile = 94.4), four times for ‘percentile > 92.9’ (interpolated percentile = 96.45), once for ‘percentile > 96.6’ (interpolated percentile = 98.3), and once for ‘percentile > 99’ (interpolated percentile = 99.5).

## 3. Results

### 3.1. Neuropsychological Findings

Table 3 shows analyses of the simplest and perhaps most commonly calculated standard score, i.e., the *z* score, obtained from *n* = 96 neurological inpatients. The median standard (*z*) scores (*Mdn*(*z*)) from all performance assessments were negative or very close to zero (−1.16 < *Mdn*(*z*) < +0.03), yet all their interquartile ranges included *z* = 0, with the exception of the RWT Lex Switch assessment. Minima and maxima indicated that we were considering broadly distributed individual standard scores that fell between −2.58 (lowest minimum) < *z* < +3.09 (highest maximum) at the extremes. Conventional *t*-tests of statistical significance (one sided, *μ*(*z*) < 0) yielded rejection of the null hypothesis in nearly all variables, indicating that the study patients performed below average performance in the standardization samples. Exceptions were WST (the indicator of *g_c_*) and RFFT ER (error rate on the figural fluency assessment), for which average performance in the standardization samples predicted the patient’s performance reasonably well.

With regard to self-reported depression on the BDI–FS, the *Mdn*(*z*) = +1.16 fell more than one standard deviation above the population average, the interquartile range did not include *z* = 0, and the distribution of individual standard scores was generally shifted toward positive values that fell within −0.15 (minimum) < *z* < +3.09 (maximum) at the extremes. A conventional *t*-test of statistical significance (one sided, *μ*(*z*) > 0) yielded rejection of the null hypothesis, indicating that the study patients reported more depressive symptoms than expected based on the average mood in the standardization sample.

### 3.2. Initial Exploration of the Correlation Matrix

Table 4 shows all Spearman rank correlations between standard scores based on observed scores (i.e., *r_xy_*; below diagonal) as well as based on true scores (i.e., *r_xtyt_*; above diagonal). Details about the calculation of the observed standard scores (*z*) can be found in the Section 2. True score correlations arise from their respective observed score correlations, plus the application of Spearman’s disattenuation correction for imperfect reliabilities, as described in the Section 2 in detail (Equation (1)). As a rule of thumb, true score correlations are always higher than their corresponding observed score correlations, due to the applied disattenuation for imperfect reliabilities.

### 3.3. Correlation Matrix in Detail: A Quick Look at Convergent Validity

#### 3.3.1. Intelligence (Italicized Black Area of the Matrix)

The observed score correlation between Raven 2 A–C and WST amounts to *r_S_* = 0.653, and the corresponding true score correlation amounts to *r_S_* = 0.727 after correction for imperfect reliabilities of the Raven 2 A–C and WST scores (see Table 4). As predicted by the CHC model of intelligence (see Introduction for details), the magnitude of these correlations provides an estimate of the convergent validity for these two measures of intelligence, with Raven 2 A–C serving as a proxy for fluid intelligence (*g_f_*) and WST serving as a proxy for crystallized intelligence (*g_c_*). Their relatively strong convergence supports the idea that broad cognitive abilities, including *g_f_* and *g_c_*, might jointly contribute to *g*.

#### 3.3.2. Executive Functioning (Italicized Grey Area of the Matrix)

Correlations between scores that were obtained from EF assessments (M-WCST, RWT, and RFFT) were also calculated. Observed score (true score in brackets) correlations between the EF scores ranged from *r_S_* = −0.008 (RFFT UD / RFFT ER; *r_S_* = −0.011, RFFT UD/RFFT ER) to *r_S_* = 0.917 (M-WCST EFc/M-WCST Persev; *r_S_* = 0.981, M-WCST EFc/M-WCST Persev). Inspection of these coefficients suggests huge heterogeneity with regard to the magnitude of the associations between EF scores. Correlations between the two independent M-WCST scores (Categ, Persev with *r_S_* = 0.700, observed score; *r_S_* = 0.753, true score) as well as those between the four RWT scores (Lex Fluency, Lex Switch, Sem Fluency, Sem Switch with *r_S_* ≥ 0.522, observed score; *r_S_* ≥ 0.647, true score) were substantial. Correlations between scores that were obtained from different tests (e.g., correlations between M-WCST and RWT scores) seemed to be weak or moderate at best (*r_S_* ≤ 0.318, observed score; *r_S_* ≤ 0.419, true score). We examine this topic in more detail below.

### 3.4. Correlation Matrix in Detail: A Quick Look at Discriminant Validity

#### 3.4.1. Discriminant Validity of Cognition against Depression (Non-Italicized Grey Area of the Matrix)

Observed score (true score in brackets) correlations between all performance scores (intelligence; EF) and self-reported depression (BDI–FS) ranged from *r_S_* = −0.302 (*r_S_* = −0.388; RWT Sem Switch) to *r_S_* = 0.096 (*r_S_* = 0.108; M-WCST Categ). Tests of statistical significance of these coefficients indicate minor (Raven 2 A–C; RWT) or negligible (WST; M-WCST; RFFT) associations between cognitive performance and self-reported depressive mood.

#### 3.4.2. Discriminant Validity of Executive Functioning against Intelligence (Non-Italicized Black Area of the Matrix)

Correlations between EF assessments (M-WCST, RWT, and RFFT) and measures of intelligence (Raven 2 as a proxy for *g_f_* and WST as a proxy for *g_c_*) were the focus of the study. Observed score (true score in brackets) correlations between EF scores and Raven 2 A–C (*g_f_*) ranged from *r_S_* = 0.237 (M-WCST Categ; *r_S_* = 0.265, M-WCST Categ) to *r_S_* = 0.451 (RFFT UD; *r_S_* = 0.566, RWT Sem Switch). Observed score (true score in brackets) correlations between EF scores and WST (*g_c_*) ranged from *r_S_* = 0.100 (M-WCST Categ; *r_S_* = 0.106, M-WCST Categ) to *r_S_* = 0.485 (RWT Lex Fluency; *r_S_* = 0.548, RWT Sem Switch). Inspection of these coefficients suggests that moderate associations may exist between indicators of EF and intelligence. We examine this topic in more detail below.

### 3.5. Reprise: Convergent Validity of Executive Functioning

Statistical significance tests were conducted that tested the equality of correlations between (verbal, figural) fluency scores (obtained from RWT and RFFT) and executive cognition scores (obtained from M-WCST) against correlations between the executive cognition scores (obtained from M-WCST), i.e., against the convergent validity coefficients of the M-WCST scores. These analyses were conducted with the online tool that is provided by *psychometrica.de* [59] for testing whether correlation coefficients differ from arbitrarily chosen values. The test is approximate, and it proceeds via Fisher-*Z*-transformation, as described by Eid et al. [60].

As previously stated, the convergent validity of M-WCST Categ (abstraction) and M-WCST Persev (cognitive flexibility) amounted to *r_S_* = 0.700 (observed score; *r_S_* = 0.753, true score). These two coefficients served as benchmarks for evaluating the convergence of the various fluency scores against these M-WCST sores (Table 5). Inspection of Table 5 reveals that *all* correlations between the fluency scores and the M-WCST scores fell below their respective coefficients of convergent validity. In addition, *all* estimates of variance shared between the M-WCST scores and the fluency scores were very low (<7% for observed scores, <10% for true scores).

These data suggest that none of the indicators of fluency examined here converge to the M-WCST scores. Hence, indicators of fluency do not seem to converge to indicators of executive cognition/cognitive flexibility, casting doubt on the unity of EF.

### 3.6. Reprise: Discriminant Validity of Executive Functioning against Intelligence

#### 3.6.1. Discriminant Validation against Convergence of Intelligence Indicators

Statistical significance tests were conducted that tested the equality of correlations between EF scores (obtained from M-WCST, RWT and RFFT) and intelligence scores against correlations between the intelligence scores (obtained from Raven 2 A–C and WST), i.e., against the convergent validity coefficients of the intelligence scores. These analyses were also conducted with the online tool that is provided by *psychometrica.de* [59] for testing whether correlation coefficients differ from arbitrarily chosen values [60].

As previously stated, the convergent validity of the Raven 2 A–C and the WST amounted to *r_S_* = 0.653 (observed score; *r_S_* = 0.727, true score). These two coefficients served as benchmarks for evaluating the divergence of the various EF scores against these intelligence sores (Table 6). Inspection of Table 6 reveals that *all* correlations between the EF scores and the intelligence scores fell below their respective coefficients of convergent validity, both for the observed scores and for the true scores.

In addition, *all* estimates of variance shared between the EF scores and the intelligence scores were low. For observed scores, M-WCST scores shared less than 12%, RWT scores shared less than 20%, and RFFT scores shared less than 21% variance with the Raven 2 A–C scores. M-WCST scores shared less than 6%, RWT scores shared less than 24%, and RFFT scores shared less than 9% variance with the WST scores. For true scores, M-WCST scores shared less than 16%, RWT scores shared less than 33%, and RFFT scores shared less than 30% variance with the Raven 2 A–C scores. M-WCST scores shared less than 7%, RWT scores shared less than 31%, and RFFT scores shared less than 12% variance with the WST scores.

#### 3.6.2. Discriminant Validity of Executive Functioning against Facets of Intelligence

A secondary research question was whether indicators of EF (here denoted as EF_x_) converged preferentially to the indicator of *g_f_* or to the indicator of *g_c_*, respectively. For example, the correlation between M-WCST EFc scores and Raven 2 A-C scores (a proxy for *g_f_*) may equal the correlation between M-WCST EFc scores and WST scores (a proxy for *g_c_*). Statistical significance tests were conducted that tested the equality of correlations between EF scores (obtained from M-WCST, RWT and RFFT) and *g_f_* against correlations between these EF scores and *g_c_* (Table 7). These analyses were conducted with the online tool that is provided by *psychometrica.de* [59].

Inspection of Table 7 reveals that the null hypothesis of equal correlations was solely rejected in one single comparison, i.e., the RFFT-based figural fluency for true scores. The overall picture here is, however, that differential correlations EF_x_—*g_f_* and EF_x_—*g_c_* were not discernible in the present study. Perhaps with the exception of productivity on figural fluency assessments, these data do not support the idea that *g_f_* might account better for variability in EF_x_ than does *g_c_*.

#### 3.6.3. Discriminant Validity of Facets of Executive Functioning against Intelligence

Another secondary research question was whether distinct indicators of EF (here denoted as EF_x_ and EF_y_, respectively) converged preferentially to the indicator of *g_f_* (or likewise to the indicator of *g_c_*). For example, the correlation between M-WCST EFc scores and Raven 2 A–C scores (a proxy for *g_f_*) may equal the correlation between RWT Lex Fluency scores and Raven 2 A–C scores. In order to keep this analysis manageable, we (a priori) selected the following three EF scores for processing: M-WCST EFc (executive cognition) scores, RWT Lex Fluency (lexical fluency) scores, and RFFT UD (figural fluency) scores. Statistical significance tests were conducted that tested the equality of correlations between an EF_x_ score and *g_._* (i.e., either *g_f_* or *g_c_*) against correlations between an EF_y_ score and *g_._* (Table 8). These analyses were conducted with the online tool that is provided by *psychometrica.de* [59].

Inspection of Table 8 reveals that the null hypothesis of equal correlations was solely rejected in two comparisons, i.e., the comparison between M-WCST-based executive cognition and RWT-based lexical fluency with regard to *g_c_* for observed scores as well as for true scores, suggesting a stronger association between lexical fluency and *g_c_* compared to executive cognition and *g_c_*. All remaining comparisons between correlations EF_x_ − *g_._* and EF_y_ − *g_._* proved non-significant. These data support a somewhat higher correlation between lexical fluency and *g_c_* compared to the correlation between executive cognition and *g_c_*. However, differential correlations between the three examined indicators of EF and *g_f_* were not discernible in the present study.

## 4. Discussion

The present data suggest that various indicators of EF can be discriminated from indicators of intelligence. Specifically, the M-WCST scores can neither be accounted for by indicators of *g_f_* nor by those of *g_c_*, given that estimates of shared variance did in no case surmount 12% for observed scores and 15% for true scores. The verbal fluency (RWT) scores can also neither be fully accounted for by indicators of *g_f_* nor by those of *g_c_*, albeit the estimates of variance shared with them were around twice as high for fluency scores as for M-WCST scores (i.e., they amounted to around 25% for observed scores and 33% for true scores). Finally, neither the figural fluency (RFFT) productivity nor error proneness seemed be fully accountable for by indicators of *g_f_* nor by those of *g_c_*, although productivity scores showed somewhat higher estimates of shared variance with *g_f_* (around 20% for observed scores and 30% for true scores) compared to *g_c_* (around 8% for observed scores and 11% for true scores).

The study results suggest that general intellectual abilities do not fully account for EF abilities. In particular, executive cognition/cognitive flexibility seems to represent a cognitive ability that is, by-and-large, independent of general intellectual abilities. Fluid and crystallized facets of intelligence are both moderately associated with verbal fluency, whereas solely fluid facets of intelligence are moderately associated with figural fluency. We conclude that EF—especially one of its core elements, i.e., executive cognition/cognitive flexibility—remains a psychometrically defendable neuropsychological construct.

We also observed good convergent validity of indicators of general intelligence (Raven’s matrices, vocabulary test), but poor convergent validity of executive cognition/cognitive flexibility (as indicated by performance on the M-WCST) against fluency indicators of EF. These data support the assumption of general intellectual abilities, but they clearly do not support the assumption of general executive abilities.

### 4.1. General Intellectual Abilities

With regard to intelligence, the good convergent validity of Raven’s matrices and a vocabulary test (*r_xtyt_* = 0.727; *R*^2^ = 0.53) is a remarkable result. Firstly, solving Raven’s matrices, mainly tagging fluid facets of intelligence, *g_f_*, and retrieving verbal knowledge, mainly tagging crystallized facets of intelligence, *g_c_*, represent quite dissimilar cognitive demands. Secondly, the solution of Raven’s matrices is a timed test, whereas the retrieval of verbal knowledge does not impose time constraints. The high correlation between these two apparently dissimilar indicators of intelligence suggests that general intellectual ability accounts for the concordant ranking on both indicators of intelligence. Spearman’s construct of general intelligence, *g* (see [9] for discussion) may be a reasonable assumption, although this topic was not the focus of the present study.

### 4.2. General Executive Abilities

With regard to EF, the poor convergent validity of executive cognition/cognitive flexibility and multiple indicators of fluency (0.087 ≤ *r_xtyt_* ≤ 0.304; 0.008 ≤ *R*^2^ ≤ 0.092) is a surprising result. Conceptually, maintaining cognitive flexibility and fluent mental productivity may be considered as resulting from similar cognitive abilities. Specifically, the presence of *generalized* perseverative tendencies should affect the ranking on both indicators of EF in similar (detrimental) ways. However, the present data did not support the assumption of generalized perseverative tendencies. We conclude that EF should be considered as a non-unitary neuropsychological construct.

Future research should address the largely unknown factorial structure of EF, preferably through latent variable modeling. These studies should examine the assumption of general (sometimes also referred to as ‘central’) executive abilities e.g., [61,62,63], which may be allocated in a single neural system (presumably a focal locus in the prefrontal cortex such as the dorsolateral prefrontal cortex, e.g., [64]). Alternatively, multiple independent executive abilities may exist, as suggested by alternative proposals [65,66,67].

### 4.3. Mental Fluency and Intelligence

Mental productivity concerning verbal fluency showed moderately strong correlations with fluid and crystallized facets of intelligence. Henry & Crawford [68] provided a meta-analysis of the sensitivity of verbal fluency to the presence of focal cortical lesions. Relative to healthy controls, participants with focal frontal injuries had large and comparable deficits in lexical and semantic fluency, which could not simply be accounted for by intellectual (dis-)abilities. The authors reported that lexical fluency was more strongly related to the presence of frontal lobe lesions than the WCST scores, and that temporal-lobe lesions were associated with a lesser deficit on lexical fluency but a larger deficit on semantic fluency. The dissociation between lexical–frontal and semantic–temporal verbal fluency was also corroborated by more recent behavior–lesion mapping studies of left hemisphere stroke patients [69,70].

Intellectual abilities may be involved in the formation of verbal retrieval strategies, such as clustering. Clustering refers to generating words within self-induced subcategories (e.g., the subcategory ‘farm animals’ from the semantic category ‘animals’), and self-induced clustering and switching between clusters may be dissociable components of verbal fluency [71]. Troyer at al. [72] reported that lexical fluency switching was impaired only in patients with frontal lobe lesions, and that semantic fluency clustering was impaired only in patients with temporal lobe lesions.

Taken together, tests of verbal fluency provide easily applicable assessment techniques, yet the nature of the cognitive processes involved, and their neural substrates, are still under debate. Fluid and crystallized intellectual abilities are both involved in a to-be-determined manner, but their variability cannot fully account for the variability observed in verbal productivity. Future studies should examine the hypothesis, which is based on Troyer et al.’s [72] finding, that the essential EF component of verbal fluency is related to the ability to switch between self-induced clusters.

Mental productivity concerning figural fluency showed moderately strong correlations with fluid, but not with crystallized, facets of intelligence. Five-point tasks are practiced much less often in daily life than are verbal retrieval tasks, such that the novelty of five-point tasks may place higher demands on structuring abilities than on retrieval abilities. Tests of figural fluency provide easily applicable assessment techniques, yet the nature of the cognitive processes involved, and their neural substrates, in figural productivity are still quite enigmatic. Fluid intellectual abilities are involved, but they cannot fully account for the variability observed in figural productivity. Future studies should examine the hypothesis that the essential EF component of figural fluency is related to the ability to switch between self-induced clusters of eligible figural designs.

### 4.4. Executive Cognition/Cognitive Flexibility and Intelligence

Some previous studies have addressed relationships between WCST-based indicators of EF and intelligence [30,31,36,37,38,39,40,73,74,75]. Notably, Jewsbury et al. [31] showed that Wisconsin perseveration errors were subsumable under CHC (cf. Introduction) broad cognitive abilities based on factor-analytic methods. Specifically, Wisconsin perseveration errors were found to be related to *g_v_* and *g_f_*, i.e., visuospatial (*g_v_*) and fluid (*g_f_*) facets of intelligence (see also [36]). A patient study also corroborated the conclusion that aspects of Wisconsin card sorting performance and fluid intelligence may be highly correlated. Notably, Roca et al. [30] compared patients who suffered from frontal lobe lesions and controls. The authors reported that the frequency of Wisconsin *total* errors no longer differed between the two groups when they were matched with regard to a widely used indicator of *g_f_* (i.e., the Culture Fair Test [76]). The authors suggested that the between-group variance in Wisconsin error proneness was negligible, once the between-group variance in fluid intelligence was accounted for. This conclusion stands in obvious conflict with Milner’s [29] assertion that measuring Wisconsin *perseveration* errors allows the detection of executive dysfunctions in patients with frontal brain lesions, in the absence of noticeable declines in intelligence.

A recent meta-analysis examined associations between WCST scores and intelligence [33]. Across all studies that were included in the meta-analysis, indicators of *g_c_* correlated at indistinguishably low levels (in terms of absolute values) with WCST categories (*r* = 0.33, confidence interval 0.26 … 0.39) and with WCST perseveration errors (*r* = −0.31, confidence interval −0.36 … −0.26). Likewise, indicators of *g_f_* correlated at indistinguishably low levels (in terms of absolute values) with WCST categories (*r* = 0.34, confidence interval 0.27 … 0.39) and with WCST perseveration errors (*r* = −0.29, confidence interval −0.34 … −0.24). As is the case in the present study, these results lead to marginal estimates of shared variance between the two cognitive domains (intelligence, executive cognition/cognitive flexibility) that lie between 8 and 12% based on these meta-analytic central-tendency estimates. The upper limits of the present results (12% for observed scores and 15% for true scores) are a little higher than the meta-analytic central-tendency estimates. The present estimates of shared variance between the two cognitive domains (intelligence, executive cognition/cognitive flexibility) are well in agreement with the meta-analytic data, when the comparison is based on the upper limits of the meta-analytic confidence-interval estimates (between 12 and 15%).

Taken together, the present data and the available evidence in the literature suggests that the usually assessed WCST-based indicators of EF (i.e., categories, perseveration errors) can be well discriminated from indicators of intelligence. This conclusion implies that the assessment of categories and perseveration errors on the WCST as indicators of EF remains a psychometrically defendable practice in clinical neuropsychology [21], which cannot be replaced by the assessment of intellectual abilities.

Our conclusion stands in contrast to the idea of a ‘multiple-demand’ system, which is supposed to subserve general intellectual abilities, i.e., *g*. The idea of a ‘multiple-demand’ system conceptualizes a general problem solver that comes into the play whenever the complexity of cognitive demands is sufficiently high [9]. However, our data do not support a unitary view of cognitive abilities. A distinction between intellectual and executive abilities seems to be in place, and WCST-based scores are among the most promising candidate indicators of EF.

### 4.5. Study Limitations

One obvious limitation of the present study lies in the trustworthiness of the reliability estimates of the test scores (cf. Table 2). Not all reliability coefficients were achievable in terms of consistency reliability, so that some estimates were in terms of test–retest reliability. Moreover, the assumption of reliability generalization may be misleading, especially when reliability estimates are transferred from studies of healthy participants to clinical samples [22]. For these reasons, the applied disattenuation (Equation (1)) may represent some degree of under- or overcorrection, respectively.

Another obvious limitation of the present study lies in the trustworthiness of the standardized scores. We argued that utilizing standardized scores is superior to utilizing raw scores (cf. Section 2.5.1). However, one problem that arises with standardized scores is that test standardization procedures are themselves not standardized. In fact, test standardization follows very heterogeneous paths, as is revealed by inspecting neuropsychological test manuals. Standardized scores of different neuropsychological tests may differ strongly in terms of their quality regarding the appropriateness of its socio-economic stratification. The assurance of quality standards concerning the standardization of neuropsychological tests is imperative. National and international neuropsychological societies should prioritize the definition of quality standards for standardized neuropsychological tests.

EF is often considered as an umbrella term for a diverse bundle of cognitive abilities. Our study was concerned with indicators of executive cognition/cognitive flexibility and mental fluency. Many more indicators of EF have been suggested in the neuropsychological [77,78,79,80,81,82] and in the cognitive [83,84] literature. Our study merely examined a strong selection among the many potential indicators of EF, and additional validity studies should envisage the broad landscape of potential indicators of EF.

Finally, although construct validity is an important building block of validity, criterion (including ecological) validity remains to be determined in future studies. Notably, well-designed studies exploring relationships between successful M-WCST performance and successful social integration, educational and professional achievement, and self-care in everyday life, are needed [85,86].

Our study participants included small subsamples of patients with and without brain diseases. Due to the small sample sizes of the subgroups of patients (see Table 1), it is not possible—nor intended—to compare their neuropsychological performance. The sole rationale for including such a large variety of patient subgroups was that the complete sample of patients should display a huge heterogeneity of individual cognitive abilities across the full ability spectrum (see Table 3). In other words, our study was not concerned with a better understanding of cognitive disorders as they occurred in these subgroups of patients, but solely with questions of test score validity when these are studied in a heterogeneous sample of individual cognitive abilities.

Our test battery included a test (Regensburger Wortflüssigkeits-Test, [46]) that assesses verbal fluency in the German language. Therefore, our findings concerning verbal fluency should be treated with some caution, because the generalizability of psychometric data concerning language-specific neuropsychological tests may be limited.

## 5. Conclusions

Our conclusions from these data are quite clear-cut. Apparently dissimilar indicators of intelligence converge on general intellectual abilities. Apparently dissimilar indicators of EF (executive cognition/cognitive flexibility, mental fluency) do not converge on general executive abilities. Executive abilities, although non-unitary, can be reasonably well distinguished from general intellectual abilities. Our conclusion that a distinction can be drawn between intellectual and executive abilities holds particularly true for executive cognition/cognitive flexibility as assessed by the Wisconsin card sorting task, thereby supporting the utilization of the EF construct in clinical neuropsychology. The reported data do not support the assumption of a ‘multiple-demand’ system, according to which the concept of general intellectual abilities represents a domain-general capacity for solving all kinds of complex cognitive problems.

## Figures and Tables

**Table 1 jcm-11-07138-t001:** Sociodemographic sample characteristics.

	*n* (%)	Age			Education			Sex ^a^	
		*M*	*SD*	Range	*M*	*SD*	Range	m	f
Complete study sample	96 (100)	42.76	14.65	18–68	14.23	2.49	8–20	35 ^a^	60 ^a^
Sub-samples									
Brain diseases	57 (59.4)								
Vascular	9 (9.4)	51.22	11.97	34–68	13.33	2.04	11–18	3	6
Autoimmune/inflammatory	38 (39.6)	38.58	13.47	21–64	14.66	2.77	8–19	14 ^a^	23 ^a^
Neurodegenerative	10 (10.4)	53.10	6.47	38–62	14.95	1.93	13–18	6	4
Other diseases	39 (40.6)								
Peripheral nervous system diseases	18 (18.7)	44.22	13.71	18–68	14.31	2.19	12–20	6	12
Non-neurological (e.g., functional) or no disease	21 (21.9)	40.52	2.19	18–67	13.43	2.27	10–19	6	15

Age and education are presented in years. ^a^ One preferred not to say.

**Table 2 jcm-11-07138-t002:** Reliability estimates of all test scores. When achievable, estimates are preferentially in terms of consistency reliability (*r*_SB_) or Cronbach’s *α*; when consistency reliability was not achievable, estimates are in terms of test–retest reliability (*r*_tt_).

Test Score	Reliability Estimate
Raven 2	
(Set) A–C	0.89
(Set) B–E	0.84
WST	0.95
M-WCST	
Executive Function Composite	0.95
Categories	0.94
Perseverative Errors	0.92
RWT	
Lexical Fluency	0.90 ^a^
Lexical Switch	0.77 ^b^
Semantic Fluency	0.85 ^b^
Semantic Switch	0.72 ^b^
RFFT	
Unique Designs	0.80
Error Ratio	0.64 ^b^
BDI–FS	0.84 ^b^

Raven 2 = Raven’s Progressive Matrices 2 Clinical Edition (German version) [41]; WST = Wortschatztest [45]; M-WCST = Modified Wisconsin Card Sorting Test [21]; EFc = Executive Function Composite; Categ = number of categories; Persev = number of perseverative errors; RWT = Regensburger Wortflüssigkeits-Test [46]; Lex Fluency = lexical fluency; Lex Switch = lexical switching; Sem Fluency = semantic fluency; Sem Switch = semantic switching; RFFT = Ruff Figural Fluency Test (German verison) [47]; UD = unique designs; ER = error ratio; BDI–FS = Beck Depression Inventory–Fast Screen (German version) [48]. Reliability estimates are displayed as Spearman–Brown split-half reliability coefficients (*r*_SB_), unless indicated otherwise. ^a^ Cronbach’s *α*
^b^ Test–retest reliability (*r*_tt_).

**Table 3 jcm-11-07138-t003:** Standard scores (*z*) from *n* = 96 neurological inpatients.

	*M*	*SD*	*Mdn*	*IQR*		*Min*	*Max*	*t* (95)	*p*
				25th	75th				
Raven 2									
A–C	−0.46	0.91	−0.61	−1.08	0.11	−2.46	2.88	−4.96	<0.001 *
B–E ^a^	−0.24	0.95	−0.28	−0.92	0.20	−2.33	2.05	−2.36 ^b^	0.010
WST	−0.01	0.74	0.03	−0.57	0.45	−1.44	1.68	−0.14	0.443
M-WCST									
EFc	−0.64	1.12	−0.57	−1.34	0.15	−2.58	1.31	−5.56	<0.001 *
Categ	−0.74	1.1	−1.00	−1.43	0.31	−2.58	1.1	−6.61	<0.001 *
Persev	−0.41	1.09	−0.50	−1.08	0.61	−2.58	1.48	−3.67	<0.001 *
RWT									
Lex Fluency	−0.54	1.07	−0.46	−1.30	0.14	−2.58	1.88	−4.99	<0.001 *
Lex Switch	−0.97	1.14	−1.16	−1.75	−0.21	−2.58	2.58	−8.32	<0.001 *
Sem Fluency	−0.41	1.13	−0.50	−1.21	0.33	−2.58	2.33	−3.56	<0.001 *
Sem Switch	−0.51	1.03	−0.38	−1.04	0.28	−2.58	1.75	−4.82	<0.001 *
RFFT									
Unique Designs	−0.35	0.96	−0.45	−0.94	0.20	−2.54	3.09	−3.54	<0.001 *
Error Ratio	−0.03	0.99	−0.25	−0.67	0.78	−2.29	2.12	−0.25	0.400
BDI–FS	1.28	0.75	1.16	0.77	1.72	−0.15	3.09	16.78	<0.001 *

Raven 2 = Raven’s Progressive Matrices 2 Clinical Edition (German version) [41]; WST = Wortschatztest [45]; M-WCST = Modified Wisconsin Card Sorting Test [21]; EFc = Executive Function Composite; Categ = number of categories; Persev = number of perseverative errors; RWT = Regensburger Wortflüssigkeits-Test [46]; Lex Fluency = lexical fluency; Lex Switch = lexical switching; Sem Fluency = semantic fluency; Sem Switch = semantic switching; RFFT = Ruff Figural Fluency Test (German verison) [47]; UD = unique designs; ER = error ratio; BDI–FS = Beck Depression Inventory–Fast Screen (German version) [48]. Student’s *t*-tests. *p*-values refer to statistical significance of H_1_ (one-sided): *μ* < 0. ^a^
*n* = 89 ^b^
*df* = 88 * *α* < $\frac{\alpha }{n\;\left(Tests\right)}$, * *α* < $\frac{0.05}{13}$ = 0.0038 (* *α* equals the significance level after Bonferroni correction based on an initial significance level of *α* < 0.05).

**Table 4 jcm-11-07138-t004:** Spearman rank correlations between *z* scores based on observed scores (i.e., *r_xy_*; below diagonal) and based on true scores (i.e., *r_xtyt_*; above diagonal).

	Raven 2 A–C	Raven 2 B–E ^a^	WST	M-WCST EFc	M-WCST Categ	M-WCST Persev	RWT Lex Fluency	RWT Lex Switch	RWT Sem Fluency	RWT Sem Switch	RFFT UD	RFFT ER	BDI–FS
Raven 2 A–C	*-*	*>0.999 ^b^*	*0.727*	0.357	0.265	0.387	0.501	0.543	0.389	0.566	0.547	0.343	−0.257
Raven 2 B–E ^a^	*0.899*	*-*	*0.693*	0.356	0.263	0.346	0.474	0.472	0.433	0.496	0.638	0.262	−0.180
WST	*0.653*	*0.637*	*-*	0.218	0.106	0.260	0.525	0.489	0.353	0.548	0.333	0.167	−0.157
M-WCST EFc	0.321	0.327	0.207	* - *	* 0.955 *	* 0.981 *	* 0.171 *	* 0.139 *	* 0.183 *	* 0.237 *	* 0.263 *	* 0.167 *	0.030
M-WCST Categ	0.237	0.241	0.100	* 0.902 *	* - *	* 0.753 *	* 0.155 *	* 0.087 *	* 0.113 *	* 0.156 *	* 0.213 *	* 0.143 *	0.108
M-WCST Persev	0.342	0.313	0.243	* 0.917 *	* 0.700 *	* - *	* 0.133 *	* 0.156 *	* 0.214 *	* 0.278 *	* 0.304 *	* 0.128 *	0.008
RWT Lex Fluency	0.438	0.424	0.485	* 0.158 *	* 0.143 *	* 0.121 *	* - *	* 0.882 *	* 0.647 *	* 0.733 *	* 0.349 *	* 0.047 *	−0.200
RWT Lex Switch	0.439	0.391	0.418	* 0.119 *	* 0.074 *	* 0.131 *	* 0.734 *	* - *	* 0.748 *	* 0.701 *	* 0.303 *	* 0.056 *	−0.271
RWT Sem Fluency	0.331	0.377	0.317	* 0.164 *	* 0.101 *	* 0.189 *	* 0.566 *	* 0.605 *	* - *	* 0.892 *	* 0.336 *	* 0.099 *	−0.342
RWT Sem Switch	0.443	0.397	0.453	* 0.196 *	* 0.128 *	* 0.226 *	* 0.590 *	* 0.522 *	* 0.698 *	* - *	* 0.419 *	* 0.044 *	−0.388
RFFT UD	0.451	0.538	0.290	* 0.229 *	* 0.185 *	* 0.261 *	* 0.296 *	* 0.238 *	* 0.277 *	* 0.318 *	* - *	* −0.011 *	−0.001
RFFT ER	0.253	0.198	0.130	* 0.130 *	* 0.111 *	* 0.098 *	* 0.036 *	* 0.039 *	* 0.073 *	* 0.030 *	* −0.008 *	* - *	0.049
BDI–FS	−0.217	−0.156	−0.140	0.027	0.096	0.007	−0.174	−0.218	−0.289	−0.302	−0.001	0.036	-

Raven 2 = Raven’s Progressive Matrices 2 Clinical Edition (German version) [41]; WST = Wortschatztest [45]; M-WCST = Modified Wisconsin Card Sorting Test [21]; EFc = Executive Function Composite; Categ = number of categories; Persev = number of perseverative errors; RWT = Regensburger Wortflüssigkeits-Test [46]; Lex Fluency = lexical fluency; Lex Switch = lexical switching; Sem Fluency = semantic fluency; Sem Switch = semantic switching; RFFT = Ruff Figural Fluency Test (German verison) [47]; UD = unique designs; ER = error ratio; BDI–FS = Beck Depression Inventory–Fast Screen (German version) [48]. Italicized areas of the matrix refer to measures of *convergent* validity (black area of the matrix, measures of intelligence; grey area of the matrix, measures of executive functioning). Non-italicized areas of the matrix refer to measures of *discriminant* validity (black area of the matrix, measures of intelligence against measures of executive functioning; grey area of the matrix, measures of cognition against measures of depression). ^a^
*n* = 89 ^b^ value truncated in response to rounding errors.

**Table 5 jcm-11-07138-t005:** Convergent validity of various fluency assessments against M-WCST scores.

Observed Scores						
Null Hypothesis	*r* (Fluency − M-WCST_x_) = *r* (M-WCST Categ − M-WCST Persev)				Shared Variance (*R*^2^)with M-WCST Score	
	M-WCST Categ		M-WCST Persev		M-WCST Categ	M-WCST Persev
	*Z*	*p*	*Z*	*p*		
fluency score						
RWT Lex Fluency	−7.011	<0.001 *	−7.227	<0.001 *	0.20	0.15
RWT Lex Switch	−7.685	<0.001 *	−7.129	<0.001 *	0.05	0.17
RWT Sem Fluency	−7.422	<0.001 *	−6.555	<0.001 *	0.10	0.36
RWT Sem Switch	−7.158	<0.001 *	−6.182	<0.001 *	0.16	0.51
RFFT UD	−6.595	<0.001 *	−5.823	<0.001 *	0.34	0.68
RFFT ER	−7.325	<0.001 *	−7.451	<0.001 *	0.12	0.10
**True Scores**						
**Null Hypothesis**	***r* (fluency − facet of Executive cognition) =** ***r* (abstraction − cognitive flexibility)**				**Shared Variance (*R*^2^) with facet of Executive cognition**	
	abstraction		cognitive flexibility		abstraction	cognitive flexibility
	*Z*	*p*	*Z*	*p*		
facet of fluency						
lexical fluency	−7.981	<0.001 *	−8.197	<0.001 *	0.24	0.18
lexical switching	−8.646	<0.001 *	−7.971	<0.001 *	0.08	0.24
semantic fluency	−8.393	<0.001 *	−7.391	<0.001 *	0.13	0.46
semantic switching	−7.971	<0.001 *	−6.734	<0.001 *	0.24	0.77
figural fluency	−7.401	<0.001 *	−6.460	<0.001 *	0.45	0.92
figural errors	−8.099	<0.001 *	−8.246	<0.001 *	0.20	0.16

M-WCST = Modified Wisconsin Card Sorting Test [21]; Categ = number of categories; Persev = number of perseverative errors; RWT = Regensburger Wortflüssigkeits-Test [46]; Lex Fluency = lexical fluency; Lex Switch = lexical switching; Sem Fluency = semantic fluency; Sem Switch = semantic switching; RFFT = Ruff Figural Fluency Test (German verison) [47]; UD = unique designs; ER = error ratio. *p*-values represent the statistical significance comparing two correlations with each other. Correlations were calculated according to Spearman’s rank-order correlations (*r_xy_*) for observed scores, and after application of Spearman’s attenuation formula (*r_xtyt_*) for true scores. * *α* < $\frac{\alpha }{n\;\left(Tests\right)}$, * *α* < $\frac{0.05}{6}$ = 0.0083 (* *α* equals the significance level after Bonferroni correction based on an initial significance level of *α* < 0.05).

**Table 6 jcm-11-07138-t006:** Discriminant validity of executive functioning (EF) assessments against measures of intelligence.

Observed Scores						
Null Hypothesis	*r* (EF Score − Measure of Intelligence) = *r* (Raven 2 A–C − WST)				Shared Variance (*R*^2^) with Intelligence Score	
	Raven 2 A–C		WST		Raven 2 A–C	WST
EF score	*Z*	*p*	*Z*	*p*		
M-WCST EFc	−4.351	<0.01 *	−5.535	<0.01 *	0.103	0.43
M-WCST Categ	−5.230	<0.01 *	−6.593	<0.01 *	0.56	0.10
M-WCST Persev	−4.124	<0.01 *	−5.169	<0.01 *	0.117	0.59
RWT Lex Fluency	−3.030	0.01 *	−2.454	0.07	0.192	0.235
RWT Lex Switch	−3.018	0.01 *	−3.266	0.01 *	0.193	0.175
RWT Sem Fluency	−4.243	<0.01 *	−4.394	<0.01 *	0.110	0.100
RWT Sem Switch	−2.970	0.01 *	−2.850	0.02 *	0.196	0.205
RFFT UD	−2.874	0.02 *	−4.681	<0.01 *	0.203	0.84
RFFT ER	−5.066	<0.01 *	−6.299	<0.01 *	0.64	0.17
**True Scores**						
**Null Hypothesis**	***r* (facet of EF − facet of intelligence) =** ***r* (*g_f_* − *g_c_*)**				**Shared Variance (*R*^2^) with** **facet of intelligence**	
	** *g_f_* **		** *g_c_* **		** *g_f_* **	** *g_c_* **
facet of EF	*Z*	*p*	*Z*	*p*		
executive cognition	−5.330	<0.01 *	−6.795	<0.01 *	0.127	0.48
abstraction	−6.314	<0.01 *	−7.905	<0.01 *	0.70	0.11
cognitive flexibility	−4.994	<0.01 *	−6.365	<0.01 *	0.150	0.68
lexical fluency	−3.621	<0.01 *	−3.307	<0.01 *	0.251	0.276
lexical switching	−3.064	0.01 *	−3.775	<0.01 *	0.295	0.239
semantic fluency	−4.972	<0.01 *	−5.374	<0.01 *	0.151	0.125
semantic switching	−2.744	0.03 *	−2.996	0.01 *	0.320	0.300
figural fluency	−3.009	0.01 *	−5.593	<0.01 *	0.299	0.111
figural errors	−5.484	<0.01 *	−7.306	<0.01 *	0.118	0.28

Raven 2 = Raven’s Progressive Matrices 2 Clinical Edition (German version) [41]; WST = Wortschatztest [45]; M-WCST = Modified Wisconsin Card Sorting Test [21]; EFc = Executive Function Composite; Categ = number of categories; Persev = number of perseverative errors; RWT = Regensburger Wortflüssigkeits-Test [46]; Lex Fluency = lexical fluency; Lex Switch = lexical switching; Sem Fluency = semantic fluency; Sem Switch = semantic switching; RFFT = Ruff Figural Fluency Test (German verison) [47]; UD = unique designs; ER = error ratio. *p*-values represent the statistical significance comparing two correlations with each other. Correlations were calculated according to Spearman’s rank-order correlations (*r_xy_*) for observed scores, and after application of Spearman’s attenuation formula (*r_xtyt_*) for true scores. * *α* < $\frac{\alpha }{n\;\left(Tests\right)}$, * *α* < $\frac{0.05}{9}$ = 0.0056 (* *α* equals the significance level after Bonferroni correction based on an initial significance level of *α* < 0.05).

**Table 7 jcm-11-07138-t007:** Are correlations between executive functioning (EF_x_) and *g_f_* equal to correlations between EF_x_ and *g_c_*?

Null Hypothesis	Observed Scores	
	*r* (EF_x_ Score − Raven 2 A–C) = *r* (EF_x_ Score − WST)	
EF score	*Z*	*p*
M-WCST EFc	1.379	0.084
M-WCST Categ	1.616	0.053
M-WCST Persev	1.210	0.113
RWT Lex Fluency	−0.629	0.265
RWT Lex Switch	0.274	0.392
RWT Sem Fluency	0.173	0.431
RWT Sem Switch	−0.132	0.447
RFFT UD	2.043	0.021
RFFT ER	1.458	0.072
**Null Hypothesis**	**True Scores**	
	***r* (facet of EF_x_ − *g_f_*) = ** ***r* (facet of EF_x_ − *g_c_*)**	
facet of EF	*Z*	*p*
executive cognition	1.910	0.028
abstraction	2.122	0.017
cognitive flexibility	1.770	0.038
lexical fluency	−0.374	0.354
lexical switching	0.845	0.199
semantic fluency	0.512	0.304
semantic switching	0.292	0.385
figural fluency	3.192	0.001 *
figural errors	2.395	0.008

Raven 2 = Raven’s Progressive Matrices 2 Clinical Edition (German version) [41]; WST = Wortschatztest [45]; M-WCST = Modified Wisconsin Card Sorting Test [21]; EFc = Executive Function Composite; Categ = number of categories; Persev = number of perseverative errors; RWT = Regensburger Wortflüssigkeits-Test [46]; Lex Fluency = lexical fluency; Lex Switch = lexical switching; Sem Fluency = semantic fluency; Sem Switch = semantic switching; RFFT = Ruff Figural Fluency Test (German verison) [47]; UD = unique designs; ER = error ratio. *p*-values represent the statistical significance comparing two correlations with each other. Correlations were calculated according to Spearman’s rank-order correlations (*r_xy_*) for observed scores, and after application of Spearman’s attenuation formula (*r_xtyt_*) for true scores. * *α* < $\frac{\alpha }{n\;\left(Tests\right)}$, * *α* < $\frac{0.05}{2}$ = 0.025 (* *α* equals the significance level after Bonferroni correction based on an initial significance level of *α* < 0.05).

**Table 8 jcm-11-07138-t008:** Are correlations between executive functioning (EF_x_) and *g_._* (i.e., either *g_f_* or *g_c_*) equal to correlations between EF_y_ and *g*?

Null Hypothesis	Observed Scores					
	*r* (M-WCST EFc − Intelligence Score) = *r* (RWT Lex Fluency − Intelligence Score)		*r* (M-WCST EFc − Intelligence Score) = *r* (RFFT UD − Intelligence Score)		*r* (RWT Lex Fluency − Intelligence Score) = *r* (RFFT UD − Intelligence Score)	
intelligence score	*Z*	*p*	*Z*	*p*	*Z*	*p*
Raven 2 A-C	−0.977	0.164	−1.137	0.128	0.123	0.451
WST	−2.294	0.011 *	−0.675	0.250	−1.787	0.037
**Null Hypothesis**	**True Scores**					
	***r* (executive cognition − facet of intelligence) =** ***r* (lexical fluency − facet of Intelligence)**		***r* (executive cognition − facet of intelligence) =** ***r* (figural fluency − facet of intelligence)**		***r* (lexical fluency − facet of intelligence) =** ***r* (figural fluency − facet of intelligence)**	
facet of intelligence	*Z*	*p*	*Z*	*p*	*Z*	*p*
*g_f_*	−1.259	0.104	−1.792	0.037	−0.488	0.313
*g_c_*	−2.603	0.005 *	−0.966	0.167	1.88	0.030

Raven 2 = Raven’s Progressive Matrices 2 Clinical Edition (German version) [41]; WST = Wortschatztest [45]; M-WCST = Modified Wisconsin Card Sorting Test [21]; EFc = Executive Function Composite; RWT = Regensburger Wortflüssigkeits-Test [46]; Lex Fluency = lexical fluency; RFFT = Ruff Figural Fluency Test (German verison) [47]; UD = unique designs. *p*-values represent the statistical significance comparing two correlations with each other. Correlations were calculated according to Spearman’s rank-order correlations (*r_xy_*) for observed scores, and after application of Spearman’s attenuation formula (*r_xtyt_*) for true scores. * *α* < $\frac{\alpha }{n\;\left(Tests\right)}$, * *α* < $\frac{0.05}{9}$ = 0.0056 (* *α* equals the significance level after Bonferroni correction based on an initial significance level of *α* < 0.05).

## Data Availability

The datasets used and/or analyzed for the study are available from the corresponding author upon reasonable request.

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
