# Peer review of "Towards the Validation of Executive Functioning Assessments: A Clinical Study"

_jcm, 2022, doi:10.3390/jcm11237138_

Round 1

Reviewer 1 Report

The article proposes the validation of the evaluation of executive functions in clinical patients. It proposes to compare, correlate and differentiate between the classical model of intelligence (crystallized and fluid) and the most sensitive neuropsychological tasks used to measure executive functions.

The authors know and manage psychometric concepts adequately, argue their use and propose new ways of analyzing test data.

From the methodological point of view, I consider that by including patients and the sample of each subgroup being small, it is not possible to compare them with each other. I think that the authors should mention in the results or discussion how the performance of these patients is.

This is an emerging need in clinical neuropsychology, since it depends on sociodemographic factors in each country where the tests are applied, which makes comparisons between studies that analyze the same components of executive functions difficult.

Finally, the article proves that executive functions differ from intellectual abilities or intelligence.

Author Response

Thank you for reviewing our manuscript with that much understanding.

With regard to your methodological comment concerning small samples of patients, we added the following paragraph to the Discussion (Section 4.5., Study Limitations), as requested by you:

“Our study participants included small subsamples of patients with and without brain diseases. Due to the small sample sizes of the subgroups of patients (see Table 1), it is not possible - nor intended - to compare their neuropsychological performance. The sole rationale for including such a large variety of patient subgroups was that the complete sample of patients should display a huge heterogeneity of individual cognitive abilities across the full ability spectrum (see Table 3). In other words, our study was not concerned with a better understanding of cognitive disorders as it occurs in these subgroups of patients, but solely with questions of test score validity when these are studied in a heterogeneous sample of individual cognitive abilities.”

Reviewer 2 Report

This manuscript is well written and interesting. However, there were several things to check.

1.     The Introduction was long and seemed redundant. Please explain briefly.

2.     P5 of ss, line 226- Are some of the tests German versions? (WCST, Verbal fluency) If so, please add information on validity and reliability. If not, the results of this study need to be discussed.

3.     P10 of 22, line 418- Table 4 What do the different colors in Table 4 indicate? Are all P values significant?

Author Response

Thank you for reviewing our manuscript with that much friendliness. With regard to your comments, we made the following changes:

  1. You are right, the Introduction is relatively long, but we do not see redundancy in our writing. The intention behind our writing is to provide kind of a didactic layout of the basic psychometric argument concerning the need to establish test score validity, and the notoriously underappreciated distinction between intellectual abilities and executive functioning served as an example. We have the impression that providing this psychometric reminder is really needed in contemporary clinical neuropsychology. For that reason, we do not wish to shorten our Introduction further.
  2. Sure, the generalizability of psychometric data concerning language-specific neuropsychological tests may be limited. This argument is particularly relevant for the verbal fluency scores (see Table 2 and Paragraph 2.3.2.2. for the available psychometric (reliability) data concerning this test, which assesses verbal fluency in German language).

We added the following paragraph to the Discussion (Section 4.5., Study Limitations):

“Our test battery included a test (Regensburger Wortflüssigkeits-Test, [46]) that assesses verbal fluency in German language. Therefore, our findings concerning verbal fluency should be treated with some caution because the generalizability of psychometric data concerning language-specific neuropsychological tests may be limited.”

  1. Concerning the different colors in Table 4, we added the following sentence to the Table Notes:
    “Italicized areas of the matrix refer to measures of convergent validity (black area of the matrix, measures of intelligence; grey area of the matrix, measures of executive functioning). Non-italicized areas of the matrix refer to measures of discriminant validity (black area of the matrix, measures of intelligence against measures of executive functioning; grey area of the matrix, measures of cognition against measures of depression).“

Are all P values significant?

We do not wish to conduct null hypothesis tests of the correlations (H0: r = 0) in Table 4.  We had no research question regarding this issue. We utilized null hypothesis tests solely in the context of the relevant research questions such as the equality of two correlations (H0: r1 = r2).